# London Chest Activity of Daily Living: Reliability and Validity of the European Portuguese Version in Heart Failure Patients

**DOI:** 10.3390/healthcare13040377

**Published:** 2025-02-10

**Authors:** Isabel J. Oliveira, Bruno M. Delgado, Cecília Mota, Inês Gomes, Pedro Lopes Ferreira

**Affiliations:** 1Fernando Pessoa University, Praça de 9 de Abril, 4249-004 Porto, Portugal; 2Center for Health Studies and Research of the University of Coimbra, Avenida Dias da Silva, 3004-512 Coimbra, Portugal; pedrof@fe.uc.pt; 3Cardiology Department, Santo António University Hospital Center, Largo Prof. Abel Salazar, 4099-001 Porto, Portugal; bruno.m.delgado@gmail.com; 4Cardiology Department of the Arrabida Local Unit, 2910-446 Setúbal, Portugal; cecilia.almeida@chs.min-saude.pt; 5Human and Social Sciences Faculty, Fernando Pessoa University, 4249-004 Porto, Portugal; igomes@ufp.edu.pt

**Keywords:** heart failure, activities of daily living, psychometrics, burden of disease

## Abstract

**Background/Objectives:** The common heart failure (HF) symptoms—dyspnea, fatigue, and edema—often prompt emergency visits. Dyspnea notably affects activities of daily living (ADLs), making its assessment crucial for evaluating therapeutic success. This study assesses the reliability and validity of the European Portuguese version of the London Chest Activity of Daily Living (LCADL) scale, originally validated in 2010, to evaluate ADL limitations in patients with HF. **Methods:** Following international guidelines for translation and cultural adaptation, 46 patients with HF from two cardiology departments were enrolled. The Six Minute Walk Test (6MWT) and the Minnesota Living with Heart Failure Questionnaire (MLHFQ) were used for construct validity. **Results:** A significant correlation was found between the results of the 6MWT and the LCADL total score (r = −0.504; *p* < 0.001) and the LCADL scale and the MLHFQ (r = 0.703; *p* < 0.001), except for the domestic activities dimension (r = 0.278; *p* = 0.062). Reliability revealed an α of 0.917. **Conclusions:** The study presents the validation of the European Portuguese version of the LCADL scale in patients with HF, emphasizing its reliability and cultural appropriateness. The LCADL scale has proven effective in assessing dyspnea-induced limitations in ADLs, and this study expands its utility by suggesting broader clinical setting applications. Future research should explore its adaptability in diverse healthcare settings, potentially enhancing personalized care strategies and patient outcomes. This work underscores the LCADL scale’s role in facilitating more targeted and effective interventions for managing ADL limitations in patients with HF, suggesting a significant impact on clinical practices and patient care management.

## 1. Introduction

Cardiovascular diseases are the world’s leading cause of death from non-communicable diseases and include conditions such as coronary heart disease, valvular heart disease, and hypertension [1]. Advances in medicine, technology, and the provision of evidence-based care have made it possible to delay the progression of these diseases and reduce mortality [2]. However, this also means that more and more patients are living with cardiovascular disease, which increases the onset of heart failure (HF).

HF is characterized by structural or functional alterations in the heart that result in increased intracardiac pressures and/or cardiac output unable to meet the body’s needs [3]. Although its incidence has stabilized in recent years, the burden of the disease shows an upward trend: people are living longer with HF [4]. The impact of the disease has repercussions on different dimensions of patients’ lives, resulting in functional limitations, poorer quality of life [5], and limitations in performing activities of daily living (ADLs) as a result of the associated symptoms [6]. In addition, significant costs are associated with re-hospitalization, decompensation episodes, and the financial impact on health systems [7].

The symptoms most frequently associated with HF include dyspnea, tiredness, and edema [5]. These are the symptoms that most often bring patients to the emergency room during an episode of decompensation [8]. Dyspnea significantly impacts daily activities in patients with HF by limiting their ability to perform basic tasks such as self-care, household chores, physical activities, and leisure. This limitation arises because dyspnea increases the effort required to complete these tasks, leading to fatigue and discomfort. As a result, patients with HF often experience reduced functional capacity and lower quality of life. The London Chest Activities of Daily Living (LCADL) scale helps quantify this impact by assessing how breathlessness interferes with specific daily activities [9]. It was developed for patients with chronic obstructive pulmonary disease (COPD). Dyspnea is a significant and frequently observed symptom in patients with HF and COPD, in whom it serves as an important indicator of disease severity and functional impairment. Consequently, systematic assessment of dyspnea in patients with HF is essential, as it allows for evaluating the therapeutic effectiveness of rehabilitation programs aimed at improving respiratory function, exercise tolerance, and overall quality of life. By monitoring changes in dyspnea over time, healthcare professionals can gain valuable insights into a patient’s response to interventions, thereby supporting a more tailored and outcome-driven approach to managing HF-related symptoms [10]. Reduced dyspnea is expected to enhance patient autonomy in performing ADLs, as alleviated breathlessness can facilitate greater functional independence and physical engagement in everyday tasks [11].

There is no specific instrument to assess limitations in performing ADLs due to dyspnea in patients with HF. Carvalho and colleagues [12] determined the validity and reproducibility of the LCADL scale in HF for Brazilian Portuguese-speaking patients. The results suggest that the LCADL scale is a simple self-administered instrument that can be used to determine limitations in ADL and that it can also be used in patients with HF.

There is no sufficiently established consensus for the process of translation and cultural adaptation of self-reported health assessment and measurement instruments [13,14], and, although debatable [15,16], there is evidence of cultural specificities in the performance of ADLs that impact the results obtained when using instruments of this nature [17,18].

In the field of HF management, the assessment of ADLs is crucial for evaluating patient outcomes and guiding treatment strategies. While the LCADL scale has been validated in various languages, its adaptation to European Portuguese is essential given its unique linguistic and cultural nuances distinct from other Portuguese variants, such as Brazilian Portuguese [19]. Differences in terminology, sentence construction, and cultural perceptions of health and daily activities can significantly impact the interpretation and effectiveness of health assessment tools in local contexts.

This study addresses a gap in the literature by focusing on the translation and cultural adaptation of the LCADL scale for European Portuguese-speaking patients with HF. It goes beyond mere linguistic translation to ensure cultural relevance, validity, and reliability across this distinct population. The process includes rigorous steps, such as ensuring conceptual equivalence, addressing cultural differences in the performance of ADLs, and expert reviews and cognitive interviews to reconcile any discrepancies and validate the instrument’s applicability. These measures are crucial to ensuring that the adapted LCADL scale is understandable and resonates with the specific cultural and healthcare context of Portugal, thus providing a reliable tool for clinicians and researchers alike.

By developing a culturally adapted version of the LCADL scale, this study contributes to the broader discourse on the importance of cultural sensitivity in clinical assessments and underscores the need for tailored tools in diverse healthcare settings.

Therefore, to further strengthen the evidence on the reliability and validity of the LCADL scale for patients with HF and develop a version in European Portuguese, this study aimed to test the reliability and validity of this version of the LCADL scale for patients with HF.

## 2. Materials and Methods

A multi-center cross-sectional study was developed in two stages. This report adheres to the Reporting of Observational Studies in Epidemiology (STROBE) for cross-sectional studies [20].

The first stage was the translation and cultural adaptation of the LCADL scale. In the second stage, the validity and reliability of the instrument were determined through its application in two clinical settings [21]. The taxonomy, terminology, and definition of health measures from the international consensus Consensus-based Standards for the Selection of Health Measurement Instruments were used as a reference for this report [22].

The translation and cultural adaptation process followed the guidelines of the Patient-Reported Outcome Consortium [14]. Authorization was requested from the authors of the LCADL instrument for its translation and cultural adaptation into European Portuguese, as well as for the other instruments used in this study.

The validation process targeted adult patients (aged 18 and over) with HF who were admitted to a cardiology department. The inclusion criteria were (a) clinical diagnosis of HF; (b) age 18 years and older; (c) no functional limitation in performing ADLs (e.g., due to neurological, orthopedic, or rheumatic causes); (d) no history of COPD; and (e) ability to understand the instructions and perform the activities/tasks measured by the instruments used. The data collection included a non-probabilistic convenience sample of patients with HF admitted to the cardiology departments of two hospital centers (one in the southern region and the other in the northern region of Portugal) between October 2022 and October 2023 who met the inclusion criteria.

For data collection, the version of the LCADL scale obtained in the first stage; a questionnaire that collected the sociodemographic and clinical characteristics of the participants (age, gender, body mass index (BMI), level of education, left ventricular function-ejection fraction (LVEF), and functional classification of HF by the New York Heart Association (NYHA)); the results from the Six Minute Walk Test (6MWT) [23]; and the Minnesota Living with Heart Failure Questionnaire^®^ (MLHFQ)—European Portuguese version [24] were used.

The LCADL scale [9], a patient-reported outcome measure, can be self-administered or administered as a patient interview. The first question asks whether patients live alone or with someone. Next, there is a set of 15 items assessed on a six-point Likert scale ranging from 0 (“I wouldn’t do it anyway”) to 5 (“Someone else does it for me”), with 1 to 4 representing degrees of breathlessness (1—”I don’t get breathless”; 2—“I get moderately breathless”; 3—”I get very breathless”; and 4—”I can’t do it anymore”). These 15 items are arranged in 4 dimensions: self-care (items 1 to 4), household activities—domestic (items 5 to 10), physical activity (items 11 and 12), and leisure activities (items 13 to 15). It has an additional multiple-choice question on patients’ overall perception of how much breathing affects ADL performance. The total score ranges from 0 to 75 points, with higher scores indicating more significant functional limitations in performing ADLs. The 6MWT test assesses functional capacity [23] by determining the maximum distance a person can cover in 6 min. The test must be carried out in a 30-meter corridor marked every 3 m, with pins at the ends to delimit the surrounding area. The MLHFQ is an instrument that measures the physical, emotional, and socioeconomic limitations imposed by HF on a patient’s quality of life [24]. It comprises 21 items that are answered on a six-point Likert scale between 0 (no limitation) and 5 (maximum limitation), referring to the last month. It has two dimensions: emotional (items 2, 3, 4, 5, 6, 7, 12, and 13) and physical (items 17, 18, 19, 20, and 21). The scores range from 0 to 105 and result from the sum of the scores obtained on each item, with higher scores implying a worse health-related quality of life.

To mitigate potential biases in data collection, researchers standardized the data collection process across sites and ensured that all data collectors were thoroughly trained in administering the LCADL tool consistently. Therefore, data collection was carried out by a rehabilitation nurse from each cardiology department specifically recruited for this study. The data collected were then analyzed to determine the validity and reliability of the LCADL scale for the Portuguese population. Descriptive statistics and correlations between scores and the clinical variables using the Spearman correlation were calculated using the IBM SPSS Statistics version 27 application. Cronbach’s α was used to determine internal consistency. A significance level of 0.05 was used. There is no established consensus for sample size calculation for the validation of instruments [25]. Therefore, as many participants as possible were enrolled during the time the study was authorized at each institution. Missing data were identified in demographic variables only. No imputations or adjustments were applied because the missing values did not affect the primary outcome measures or statistical analyses. Descriptive statistics were calculated on the basis of available demographic data. Sensitivity analyses were not performed, as missing data were limited to demographic variables and did not influence the primary outcomes or statistical analyses. The study’s results are thus based solely on the available data for the primary variables.

This study was authorized by the ethics committees of both participating institutions: CHS No. 026/2022F and CHUdSA No. 2022.271. All legal and ethical procedures for obtaining explicit, free, informed written consent from the participants were followed. All participants gave their explicit written consent.

## 3. Results

The translation and cultural adaptation process was carried out as planned. Two independent translators translated the original version into European Portuguese. The principal investigator reconciled these two versions, resulting in a consensus version. In the process of translating and culturally adapting the LCADL scale to European Portuguese, particular attention was paid to maintaining conceptual equivalence across linguistic versions. Discrepancies between the original and translated versions were meticulously resolved through a collaborative approach. Each discrepancy was discussed in detail by the principal researcher with healthcare professionals and language experts to ensure that the translations were linguistically accurate and culturally congruent. For instance, when differences in medical terminology arose, clinical guidelines and existing literature were consulted to decide on the most appropriate terms that reflected current clinical practice in Portugal. In cases of idiomatic expressions related to daily activities, the researchers sought input from native speakers to find phrases that conveyed similar connotations in the cultural context of the target audience. These adjustments were then rigorously tested in subsequent back-translation to verify that the original meaning was preserved. This iterative process helped to refine the translation and enhance the instrument’s reliability and applicability for Portuguese-speaking patients with HF. The back-translated version was compared with the original version, resulting in equivalent versions.

Subsequent comparative analysis between the European Portuguese and Brazilian Portuguese versions highlighted significant lexical differences, such as the use of ‘enxugar-se’ in Brazilian Portuguese versus ‘secar o corpo’ in European Portuguese. Such variations are crucial in ensuring that the instrument is culturally and linguistically appropriate for the intended population. Differences in sentence construction were also noted, with phrases like ‘…tell us how much shortness of breath you are feeling…’ in Brazilian Portuguese altered to ‘…tell us how difficult it is to breathe…’ in European Portuguese to better align with local usage and comprehension. Given the vocabulary and syntax differences between the two variants of Portuguese, careful attention was paid to ensuring that the adaptation not only translated but also transcreated the instrument to fit the cultural context of Portugal.

The European Portuguese version was then presented and discussed with five rehabilitation nurses and cardiac rehabilitation experts who provided feedback on form and expressions without suggesting changes to the content. Their expertise ensured that the instrument remained both clinically relevant and culturally adapted. Cognitive interviews were conducted in person with three patients with HF who met the study’s inclusion criteria. These interviews assessed how participants processed and understood the questions in the European Portuguese version of the LCADL scale. During these sessions, the participants were asked to articulate their thoughts and comprehension of each item, allowing for an in-depth understanding of the instrument’s clarity and cultural relevance. No modifications were made to the instrument following these interviews, indicating that the initial adaptations were well-received and understood by the target population. In the final version, the result of the consensus process of the two translations was used to validate the instrument in a clinical context. The approach to addressing these linguistic variations was informed by scholarly work on cognitive and social linguistics of Portuguese variants [19].

For validation, 46 patients were enrolled in this study, with 31 (67.40%) from the south and 15 (32.60%) from the northern region; 33 (71.70%) were male, and the mean age was 64.20 ± 13.78 years (range 29–88). Table 1 summarizes the participants’ sociodemographic and clinical characteristics.

The results from the LCADL scale and MLHFQ are presented in Table 2.

Most participants reported not living alone (*n* = 38; 84.40%) (data missing for one participant). Regarding the participants’ overall perception of how breathing impacts ADL performance, 25 participants (55.60%) responded “a little”, 4 (8.90%) “a lot”, and 16 (35.60%) “not at all” (data missing for one participant). When analyzing correlations, a statistically significant negative correlation was found between the results of the 6MWT and the LCADL total score (r = −0.504; *p* < 0.001).

A statistically significant correlation was also found between the LCADL scale and the MLHFQ (Table 3), except for the domestic activities dimension.

Table 4 presents the LCADL data stratified (mean ± SD) according to NYHA Functional Classification. After identifying significant differences in LCADL scores across NYHA functional classifications using the Kruskal–Wallis test, post hoc pairwise comparisons using the Dwass–Steel–Critchlow–Fligner test were conducted to determine specific group differences. Statistically significant differences were found between participants with NYHA classifications I, II, and III (*p* < 0.05), with more significant differences between participants with NYHA classifications I and III (*p* < 0.001).

A box plot representation was employed to visually assess the distribution of LCADL total scores across different NYHA classifications and to identify any outliers or patterns within the data. This facilitated a clear comparative analysis of the median, interquartile ranges, and overall distribution of scores among patients with HF categorized into NYHA Classes I, II, and III. Such visualization aids in evaluating the sensitivity of the LCADL scale to varying severities of HF symptoms, which is crucial for its validation in a clinical setting (Figure 1). Additionally, the box plot highlights specific outliers, marked as cases ‘1’ and ‘32’, representing individual scores that deviate significantly from the general trend within their respective NYHA classifications. Additionally, the box plot highlights specific outliers, marked as cases ‘1’ and ‘32’, which exhibit LCADL scores that deviate significantly from the general trend within their respective NYHA classifications. Notably, these participants also reported higher MLHFQ scores of 72 and 70, respectively, indicating a poorer quality of life. This correlation suggests that their significant limitations in daily activities might be closely linked with their overall health status and symptom severity.

No correlation was found between the LCADL total score and LVEF (r = −0.188; *p* = 0.210), BMI (r = −0.220; *p* = 0.150), or age (r = −0.178; *p* = 0.236).

Analyzing the reliability of the LCADL scale revealed a Cronbach’s α value of 0.917.

## 4. Discussion

The use of the LCADL scale to assess dyspnea when performing ADLs in patients with HF proved valid and reliable, consistent with findings from the Brazilian adaptation, which validated the LCADL scale under similar clinical conditions [12]. The results confirm the LCADL scale’s ability to capture the impact of HF on ADLs, similar to the observations in the Brazilian study. Although the MLHFQ results indicate that the patients in this study had fewer physical, emotional, and socio-economic limitations compared with its validation in Brazilian Portuguese [12] (31.57 vs. 50.20), statistically significant positive correlations were found between the LCADL scale and all dimensions of the MLHFQ, except for domestic activities. This suggests that, as with the Brazilian adaptation, higher levels of dyspnea are associated with a poorer quality of life, underscoring the LCADL scale’s sensitivity to the multifaceted impact of dyspnea in HF.

This study also highlighted cultural differences in symptom reporting, especially in the domain of domestic activities where correlations were weaker, possibly reflecting differing cultural perceptions of daily tasks and responsibilities. Unlike the Brazilian study, which did not find significant differentiation between NYHA classes II and III, these findings showed notable differences in LCADL scores between these classes. This divergence might suggest that symptoms of dyspnea and their impact on daily activities are perceived and reported differently in European versus Brazilian contexts.

Furthermore, the strong reproducibility of the LCADL scale in this study, as well as in the Brazilian adaptation, supports the scale’s utility for repeated measures and consistent application in clinical settings across different cultures.

These insights validate the broad applicability of the LCADL scale and highlight the need for ongoing cultural adaptations to ensure the tool accurately reflects the specific contexts of its use. The variations in how dyspnea impacts daily living across different cultures underscore the importance of considering such factors when interpreting quality of life measurements, like those obtained from the MLHFQ. The MLHFQ measures quality of life, suggesting that higher levels of dyspnea are associated with poorer quality of life. The fact that most participants in this study were male could have potentially influenced the lack of correlation between the domestic activities dimension of the LCADL scale and the MLHFQ. Gender roles and cultural expectations often lead to differences in domestic activities typically performed by men and women [26]. This could affect how male participants perceive and report their limitations in these activities. In many cultures, men may be less involved in domestic tasks such as cooking, cleaning, or other household chores, which could result in lower reported limitations in the domestic activities dimension of the LCADL scale. If male participants do not routinely perform these activities, they may not experience or perceive significant dyspnea during these tasks, leading to lower scores that do not correlate strongly with the overall quality of life as assessed by the MLHFQ. On the other hand, men may underreport difficulties in domestic activities due to societal norms or personal perceptions, potentially skewing the results [27]. If male participants do not typically engage in these activities, they might not consider them relevant to their daily functioning, even if they experience significant dyspnea in other contexts.

Variability in task performance may also explain the lack of correlation. The physical environment and personal living situations can influence the performance of domestic tasks. For instance, individuals in multi-story homes may face more challenges than those in single-level dwellings. Additionally, the diversity in living arrangements and the availability of support, such as family members or hired help, could explain the variability in reported difficulties with domestic activities.

Age may also impact the ability to perform domestic tasks, as older adults often experience physical declines [28] that can exacerbate the effects of dyspnea, even if the results obtained do not suggest it. Analyzing how age-related changes in mobility, strength, and endurance influence LCADL responses could provide valuable insights. Socioeconomic status (SES) can influence an individual’s access to resources that alleviate the burden of domestic activities. Participants with higher SES might not perceive these activities as challenging if they are accustomed to delegating such tasks, contributing to the lower correlation in self-reported limitations due to dyspnea.

Psychological conditions such as depression or anxiety might affect how participants perceive and report limitations in domestic activities. The interplay between psychological well-being and self-reported physical limitations warrants further investigation, as it may influence the responses on the LCADL scale, especially in the domestic activities dimension. Furthermore, the current items related to domestic activities on the LCADL scale may not adequately capture the variety of tasks or the specific challenges posed by dyspnea in these activities. A review and potential revision of these items could enhance the instrument’s sensitivity and specificity, ensuring that it better reflects the range of experiences of those with HF.

The lack of correlation may also reflect the differing scopes and sensitivities of the two instruments: the MLHFQ captures broader life impacts, while the LCADL scale focuses narrowly on dyspnea during specific activities. This divergence underscores the importance of using multiple tools to fully capture the varied impacts of HF on patients’ lives.

Comparing the differences in the application of the LCADL scale and the different classes of the NYHA classification, the results differ from those found by Carvalho and colleagues [12]. The participants were younger (50 ± 9) and had more significant limitations in ADLs as measured by the LCADL scale (35.9 ± 21.9). These differences in the characteristics of the participants may explain these differences.

In another study carried out later, which sought to assess the applicability of the LCADL scale in patients with HF [29], the 6MWT was used as a comparison measure, and correlations were sought with the LVEF and BMI values. The LCADL scale correlates negatively with 6MWT scores, indicating that as functional capacity decreases (as indicated by a shorter walking distance), limitations in daily activities due to dyspnea increase. However, the LCADL scale does not correlate with certain clinical measures like LVEF, BMI, or age, indicating that it is a specific measure of dyspnea’s impact rather than a general health assessment tool. However, the results differ in their association with the value of LVEF. In the study by Valadares and colleagues [29], a relationship was found between these two variables. It should be noted that the sample size in this study was 10 participants, and only patients with NYHA classification III and IV were included, making it more difficult to compare results.

As for reliability, the internal consistency (Cronbach α) of the European Portuguese Version is lower than the one found by Carvalho and colleagues (α = 0.99) and the original version in English (α = 0.98) [9,12]. However, the internal consistency value can be considered reliable because it shows an adequate interrelationship between the instruments’ items [30].

This study has limitations, primarily in sample size and demographic composition, which may affect the generalizability of findings, especially regarding gender differences in the impact of dyspnea on ADLs. The relatively small, predominantly male sample may not fully capture gender-specific variations, suggesting a need for further research with larger, more diverse cohorts. Such studies should aim to achieve a balanced gender distribution and focus on longitudinal analyses to improve understanding of the chronic progression of HF and its impact on ADLs across different patient groups. Additionally, future research should consider interventions targeting diverse populations to validate and broaden the applicability of the LCADL scale, ensuring that it meets the varied needs of patients with HF in different cultural and demographic contexts. Furthermore, the lack of correlation between the LCADL scale’s domestic activities dimension and the MLHFQ may indicate gender-specific influences on activity limitations due to dyspnea, which remains underexplored in HF research. Longitudinal studies with these considerations could further evaluate the instrument’s sensitivity to changes over time and deepen understanding of gender-related effects on quality of life in patients with HF. Future research could also benefit from a multi-instrument approach to further validate the LCADL scale across different facets of health and extend our understanding of its applicability and robustness in diverse clinical contexts.

## 5. Conclusions

This study confirms the validity and reliability of the European Portuguese version of the LCADL scale for assessing dyspnea in patients with HF. The results indicate that the LCADL scale is a robust tool for assessing the impact of dyspnea on ADLs in European Portuguese-speaking patients with HF. The LCADL scores were significantly correlated with functional capacity as measured by the 6MWT and the MLHFQ. This underscores its utility in clinical settings, particularly in assessing the effectiveness of interventions such as cardiac rehabilitation programs. However, it is important to note that the LCADL scale did not correlate with LVEF, BMI, or age, suggesting that the scale measures the functional impact of dyspnea rather than being influenced by these variables. The high Cronbach’s α value indicates excellent internal consistency, confirming the reliability of the LCADL scale in this population.

The practical implications of these findings are significant. Clinicians can confidently employ the European Portuguese LCADL scale to assess baseline dyspnea impairment and to monitor patient progress over time in response to treatment. Moreover, the successful adaptation of the LCADL scale from its original version to fit the cultural context of Portuguese-speaking patients illustrates the value of culturally adapting validated instruments rather than developing new ones. This approach saves resources and allows for quicker implementation in clinical practice.

Future research should build on these findings by exploring the use of the LCADL scale in longitudinal studies and in diverse populations to validate its broader applicability and to refine HF management strategies further. Such studies could enhance the understanding of dyspnea’s impact on quality of life and inform more targeted and effective interventions.

## Figures and Tables

**Figure 1 healthcare-13-00377-f001:**
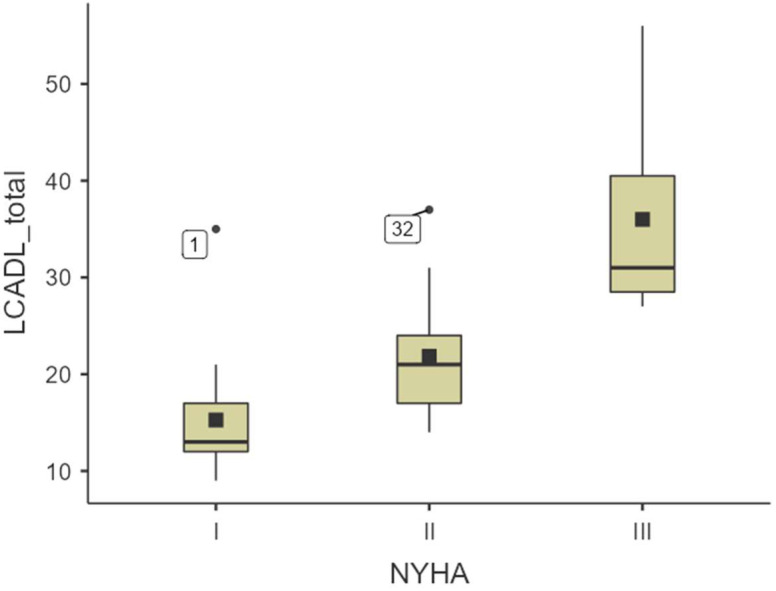
Box plot showing the distribution of LCADL total scores across different NYHA classifications (I, II, III) in patients with HF. Outliers are indicated by points labeled with their respective case numbers. The plot illustrates the variation in ADL limitations by severity of HF symptoms.

**Table 1 healthcare-13-00377-t001:** Participants’ sociodemographic and clinical characteristics.

BMI, mean ± SD (range) ^¥^	27.70 ± 4.56 (19.80–37.70)
LVEF%, mean ± SD (range)	38.90 ± 12.80 (11.00–72.00)
6MWT, mean ± SD (range)	399.00 ± 99.61 (199.00–587.00)
Education, *n* (%) ^Ɨ^	
Basic	31 (68.89)
Secondary	10 (22.22)
Higher	4 (8.89)
NYHA Functional Classification, *n* (%)	
I	18 (39.10)
II	21 (45.70)
III	7 (15.20)

^Ɨ^ Data missing for 1 participant, ^¥^ Data missing for 2 participants.

**Table 2 healthcare-13-00377-t002:** Results from LCADL scale and MLHFQ.

LCADL	Dimensions	Mean ± SD (Range)
	Self-Care	6.15 ± 3.33 (2.00–19.00)
	Domestic	6.64 ± 7.10 (0.00–29.00)
	Physical Activity	3.72 ± 1.87 (2.00–9.00)
	Leisure	4.22 ± 2.34 (0.00–13.00)
	**Total**	20.67 ± 10.94 (7.00–56.00)
**MLHFQ**		
	Physical	12.70 ± 11.67 (0.00–39.00)
	Emotional	7.74 ± 7.38 (0.00–24.00)
	**Total**	31.57 ± 27.12 (0.00–100.00)

**Table 3 healthcare-13-00377-t003:** Correlation of the LCADL scale and MLHFQ.

MLHFQ	LCADL	*r*	*p*
Physical	Self-Care	0.746	<0.001
	Domestic	0.203	0.176
	Physical activities	0.633	<0.001
	Leisure	0.534	<0.001
	Total	0.667	<0.001
Emotional	Self-Care	0.719	<0.001
	Domestic	0.270	0.070
	Physical activities	0.560	<0.001
	Leisure	0.498	<0.001
	Total	0.643	<0.001
Total	Self-Care	0.729	<0.001
	Domestic	0.278	0.062
	Physical activities	0.673	<0.001
	Leisure	0.502	<0.001
	Total	0.703	<0.001

**Table 4 healthcare-13-00377-t004:** Data stratified according to NYHA Functional Classification.

NYHA	LCADL Total	Self-Care	Domestics	Physical Activities	Leisure
**I**	14.56 ± 6.21	4.22 ± 0.81	4.28 ± 3.36	2.83 ± 1.65	3.22 ± 1.44
**II**	20.81 ± 8.88	6.86 ± 4.14	6.00 ± 6.40	4.00 ± 1.82	4.29 ± 2.10
**III**	36.00 ± 11.83	9.00 ± 1.41	15.29 ± 10.19	5.14 ± 1.57	6.57 ± 3.31

## Data Availability

The article’s data will be shared on reasonable request to the corresponding author. The raw data supporting the conclusions of this article will be made available by the authors on request.

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
