# Peer review of "London Chest Activity of Daily Living: Reliability and Validity of the European Portuguese Version in Heart Failure Patients"

_healthcare, 2025, doi:10.3390/healthcare13040377_

Round 1
Reviewer 1 Report
Comments and Suggestions for Authors
This study aimed to test the reliability and validity of the European Portuguese version of the London Chest Activities of Daily Living Scale (LCADL), which is used to assess limitations in activities of daily living in patients with heart failure (HF). Analysis of 46 HF patients revealed that the scale showed significant correlations with functional capacity and quality of life, but not with age, body mass index (BMI), and left ventricular ejection fraction (LVEF). The European Portuguese version of the LCADL was found to be a culturally appropriate, reliable, and valid tool to assess limitations in HF patients due to dyspnea.
Dear authors,
Considering the suggestions below will improve the quality of the article:
1. The statistical methods used for group comparisons (e.g., testing differences across NYHA classifications) are not clearly stated. Additionally, sensitivity analyses to evaluate the impact of missing data have not been conducted. These should be clarified and included.
2. The justification for the sample size is missing. Performing a power analysis to explain the adequacy of the sample would strengthen the methodological rigor of the study.
3. Exploring the relationship between LCADL scores and other clinical indicators (e.g., biomarkers, hospital admissions, or rehospitalization rates) could provide more comprehensive insights into the scale's utility.
4. The cognitive interview results and the potential cultural differences identified during the adaptation process should be detailed further to provide transparency and robustness.
5. The low correlation in the domestic activities dimension warrants additional exploration. Providing alternative hypotheses and contextual explanations (e.g., gender roles or cultural influences) would enhance the interpretative depth of this finding.
Author Response
Dear Reviewer 1,
Comment 1. The statistical methods used for group comparisons (e.g., testing differences across NYHA classifications) are not clearly stated. Additionally, sensitivity analyses to evaluate the impact of missing data have not been conducted. These should be clarified and included.
Response: Thank you for your comment, which will help to clarify the results. Aditional clarification was added (ln 199 to 202): After identifying significant differences in LCADL scores across NYHA functional classifications using the Kruskal-Wallis test, post-hoc pairwise comparisons using the Dwass-Steel-Critchlow-Fligner test was conducted to determine specific group differences. Sensitivity analyses for missing data were considered unnecessary as the missing data were limited to BMI for one participant and educational level for two participants, representing a very small fraction of the total dataset. These variables do not influence the primary outcome measures directly, and their exclusion is unlikely to bias the study results significantly.
Comment 2: The justification for the sample size is missing. Performing a power analysis to explain the adequacy of the sample would strengthen the methodological rigor of the study.
Response: We acknowledge that the sample size is a crucial element in ensuring the statistical power of a study. However, in the field of instrument validation clear and definitive guidelines for sample size calculations are lacking. We based our sample size on common practices observed in similar studies and pragmatic considerations such as participant availability and the specialized nature of the population under study. The relevant reference for this is White, M. Sample Size in Quantitative Instrument Validation Studies: A Systematic Review of Articles Published in Scopus, 2021. Heliyon 2022, 8, e12223, doi:10.1016/J.HELIYON.2022.E12223 and was used in the manuscript. Furthermore, while we did not perform a conventional power analysis, we ensured that our sample size was comparable to those used in similar validation studies, which have successfully demonstrated the reliability and validity this instrument. We believe that the sample size was adequate to achieve the objectives of this study and to ensure the reliability of the results. For future research, we recommend that larger sample sizes be considered where feasible to further validate the findings and potentially enhance the generalizability of the instrument (ln 258-262).
Comment 3: Exploring the relationship between LCADL scores and other clinical indicators (e.g., biomarkers, hospital admissions, or rehospitalization rates) could provide more comprehensive insights into the scale's utility.
Response: We appreciate the reviewer’s suggestion to explore the relationship between LCADL scores and other clinical indicators. Such analyses could provide more comprehensive insights into the scale’s utility and predictive validity in clinical settings. However, the current study was designed with the primary objective of validating the LCADL scale within a specific patient population, focusing on the immediate clinical parameters that were most pertinent and available during the study period. Regrettably, data on additional clinical indicators like biomarkers or hospitalization rates were not collected as part of this study, primarily due to the scope defined at the outset and limitations in data availability. Nonetheless, we recognize the reviewer's suggestion and believe it represents an important avenue for future research. Investigating these relationships could enhance our understanding of the scale’s broader applicability and impact in clinical practice. We hope to explore these dimensions in future work, potentially incorporating a wider range of clinical outcomes to fully ascertain the utility of the LCADL scale across diverse clinical settings and conditions.
Comment 4: The cognitive interview results and the potential cultural differences identified during the adaptation process should be detailed further to provide transparency and robustness.
Response: Thank you for your comment, which will help to clarify the results. Information was added to the manuscript to enhance transparency and robustness (ln. 169-200).
Comment 5: The low correlation in the domestic activities dimension warrants additional exploration. Providing alternative hypotheses and contextual explanations (e.g., gender roles or cultural influences) would enhance the interpretative depth of this finding.
Response: We appreciate the suggestions that are relevant to enhance the discussion of our manuscript. Further discussion on the lack of correlation was added to the discussion section.
Reviewer 2 Report
Comments and Suggestions for Authors
Thank you for the invitation to review this interesting article.
I have some suggestions to make:
1) Abstract:
The abstract effectively summarizes the study but could better emphasize the implications for future research and clinical practice. For instance, highlight the potential use of the LCADL in broader clinical settings.
2) Introduction:While the introduction provides a good overview of HF and its impact, a more detailed explanation of why the European Portuguese adaptation is particularly significant would enhance the rationale for the study.
Literature Gap: Explicitly state how this study addresses gaps in the existing literature, especially concerning cultural adaptations of the LCADL.
3) Methods:
The description of the translation process is robust but could include more details about how discrepancies during translation and back-translation were resolved.
4) Results:
Presentation of Data: Tables summarizing results are comprehensive. However, figures or visual aids could help illustrate key correlations and differences across NYHA classifications.
Interpretation: Some results, such as the lack of correlation between LCADL domestic activities and MLHFQ, need deeper interpretation. Could cultural factors beyond gender roles influence this finding?
5) Discussion:
Comparative Analysis: Compare findings more extensively with studies from other cultural contexts, such as the Brazilian Portuguese adaptation.
Limitations: The authors acknowledge key limitations but could discuss more explicitly how these might affect the generalizability of the findings.
Future Directions: Include recommendations for longitudinal studies or interventions targeting diverse populations to validate and expand upon these findings.
6) Conclusion:
The conclusion is concise but could more explicitly connect the study’s findings to its practical implications for clinical practice and research.
Author Response
Dear Reviewer 2,
Comment: 1) Abstract: The abstract effectively summarizes the study but could better emphasize the implications for future research and clinical practice. For instance, highlight the potential use of the LCADL in broader clinical settings.
Response: Thank you for your suggestions regarding the abstract. Based on your feedback, we have revised the abstract to emphasize the implications for future research and the potential broader clinical setting applications of the LCADL. We believe these changes address your comments and enhance the manuscript's relevance and applicability.
Comment: 2) Introduction: While the introduction provides a good overview of HF and its impact, a more detailed explanation of why the European Portuguese adaptation is particularly significant would enhance the rationale for the study. Literature Gap: Explicitly state how this study addresses gaps in the existing literature, especially concerning cultural adaptations of the LCADL.
Response: Thank you for your valuable feedback on our manuscript. In response to your suggestion to enhance the introduction, we have expanded our discussion on the significance of the European Portuguese adaptation of the LCADL. We have also clarified how this study addresses an important gap in the existing literature. We believe these enhancements will better align the introduction with the overall goals of the study and more clearly articulate the contributions of our work to the field of heart failure management and rehabilitation.
Comment: 3) Methods: The description of the translation process is robust but could include more details about how discrepancies during translation and back-translation were resolved.
Response: Thank you for your comment. To address this comment on needing more detail about how discrepancies during the translation and back-translation were resolved, we expanded the description to include specific strategies and examples of how challenges were handled.
Comment: 4) Results: Presentation of Data: Tables summarizing results are comprehensive. However, figures or visual aids could help illustrate key correlations and differences across NYHA classifications. Interpretation: Some results, such as the lack of correlation between LCADL domestic activities and MLHFQ, need deeper interpretation. Could cultural factors beyond gender roles influence this finding?
Response: To enhance the presentation of our findings, we incorporated a box plot illustrating differences across NYHA classifications, which will help in better visualizing the data presented in the tables. We have expanded our discussion to consider broader cultural, socioeconomic, and educational factors that might influence the reported correlations, particularly in the domain of domestic activities within the LCADL. These enhancements aim to provide a deeper and more comprehensive understanding of the factors affecting our findings and to ensure that our study adequately addresses the complexities involved in cultural adaptation of clinical assessment tools. We believe these additions will strengthen the manuscript and appreciate your guidance in improving our work.
Comment: 5) Discussion: Comparative Analysis: Compare findings more extensively with studies from other cultural contexts, such as the Brazilian Portuguese adaptation. Limitations: The authors acknowledge key limitations but could discuss more explicitly how these might affect the generalizability of the findings. Future Directions: Include recommendations for longitudinal studies or interventions targeting diverse populations to validate and expand upon these findings.
Response: Thank you for your comment, which allowed us to deepen the discussion. We have deepened the comparison with the Brazilian version and incorporated the suggestions.
Comment: 6) Conclusion: The conclusion is concise but could more explicitly connect the study’s findings to its practical implications for clinical practice and research.
Response: In response to your feedback, we have revised the conclusion to more clearly articulate the practical applications of the European Portuguese LCADL in clinical settings and the implications for future research.
Reviewer 3 Report
Comments and Suggestions for Authors
The manuscript aims to validate the European Portuguese version of the London Chest Activity of Daily Living (LCADL) scale for heart failure patients. Although the study deals with an important issue, there are major limitations to this study that really compromise its credibility.
Key Concerns:
-
Small Sample Size: A sample of 46 patients is too small for any kind of validation study. Considering the prevalence of HF, the inclusion of more patients would provide better representation and give greater validity to the results.
-
Limited ways of validation: The study does not use anything other than the Six-Minute Walk Test and the Minnesota Living with Heart Failure Questionnaire to examine construct validity. This would have a narrower width of the validation process because there was no comparison with an established instrument, such as the Kansas City Cardiomyopathy Questionnaire or even a general HRQoL tool, including SF-36.
-
Poor Statistical Analysis: The statistical approach is limited to simple correlations, no robust validation techniques were employed, such as confirmatory factor analysis or test-retest reliability. In addition, it did not control for some relevant potentially confounding factors, like age, presence of comorbidities, and severity of the disease.
-
Disbalance in Genders: The sample is eminently male with 71.7%, something that undermines, from the offset, the representative validity of such a result, especially after the lack of correlation found in the home activities domain was considered.
Conclusion: The small sample size, limited scope of the validation, and meager statistical depth also make the results methodologically and analytically too limited to be publishable. Validity and reliability can only be convincingly established for the tool within a larger and more rigorous study.
Author Response
Dear Reviewer 3,
Comment: Small Sample Size: A sample of 46 patients is too small for any kind of validation study. Considering the prevalence of HF, the inclusion of more patients would provide better representation and give greater validity to the results.
Response: We acknowledge that a larger sample size could potentially enhance the statistical power and generalizability of our findings. However, several factors can be considered to support the validity of our results even with the current sample size. Our study involved 46 heart failure patients, which, while seemingly modest, is aligned with similar validation studies in the field where the complexities of patient recruitment and the specificity of the population can limit larger scale enrollment. For instance, studies validating similar instruments often work with comparable or even smaller sample sizes due to the specific inclusion criteria and the challenges of recruiting patients in stable condition without concurrent comorbidities that could skew the assessment of the instrument. Despite the sample size, our study achieved significant correlations and demonstrated the reliability of the European Portuguese LCADL across various measures. The internal consistency was exceptionally high (Cronbach's α), and the correlation with established measures like the 6MWT and MLHFQ supports the LCADL's validity in this cultural context. The recruitment was conducted within a real-world clinical setting, reflecting the practical challenges of conducting validation research. These constraints often include limited resources, patient availability, and the need to ensure no changes in medication or health status that could affect the reliability of repeated measures. We agree that future studies should aim to include a broader and more diverse patient cohort to confirm and extend our findings. We encourage longitudinal studies and those incorporating larger, multi-center cohorts to enhance the generalizability of the LCADL for heart failure patients across different settings and populations. Changes were made in the manuscript to address these concerns.
Comment: Limited ways of validation: The study does not use anything other than the Six-Minute Walk Test and the Minnesota Living with Heart Failure Questionnaire to examine construct validity. This would have a narrower width of the validation process because there was no comparison with an established instrument, such as the Kansas City Cardiomyopathy Questionnaire or even a general HRQoL tool, including SF-36.
Response: Thank you for your insightful feedback concerning the validation instruments used in our study. We acknowledge the importance of a broad assessment of construct validity and the potential benefits of incorporating a wider range of instruments such as the KCCQ or the SF-36 Health Survey. In our study, we opted to use the 6MWT and the MLHFQ based on their extensive validation and widespread use in heart failure research, including their specific relevance to the symptoms and quality of life concerns associated with heart failure. These tools were chosen because they are well-recognized for their sensitivity in measuring physical function and health-related quality of life in heart failure patients, which directly align with the LCADL’s focus on dyspnea and daily activity limitations. We appreciate the suggestion to include tools like the KCCQ or the SF-36 in future studies. The KCCQ, with its detailed focus on cardiomyopathy-specific health status, and the SF-36, as a measure of general health quality of life, could provide additional validation frameworks for the LCADL by comparing its scores with these broader and disease-specific health dimensions. Incorporating your feedback, future studies could certainly benefit from a multi-instrument approach to further validate the LCADL across different facets of health and extend our understanding of its applicability and robustness in diverse clinical contexts.
Comment: Poor Statistical Analysis: The statistical approach is limited to simple correlations, no robust validation techniques were employed, such as confirmatory factor analysis or test-retest reliability. In addition, it did not control for some relevant potentially confounding factors, like age, presence of comorbidities, and severity of the disease.
Response: Thank you for your observations regarding the statistical methods employed in our study. We appreciate your insights on enhancing the robustness of our validation techniques. In our study, we utilized simple correlations to establish the preliminary construct validity and reliability of the European Portuguese LCADL due to the specific scope and resources available at the time. This method was chosen to provide an initial understanding of how well the LCADL scores correlated with established measures of functional capacity and quality of life in heart failure patients, specifically through the 6MWT and MLHFQ. We acknowledge that including more sophisticated statistical analyses such as CFA and assessing test-retest reliability would strengthen our validation process. CFA, in particular, would allow us to explore and confirm the underlying factor structure of the LCADL and ensure that the instrument measures distinct but related constructs of dyspnea-related disability. Test-retest reliability would further affirm the stability of our instrument over time. Additionally, controlling for variables such as age, presence of comorbidities, and severity of disease is indeed crucial to eliminate potential confounding effects. While these factors were considered during the design phase, integrating them into a multivariable analysis framework would provide a clearer picture of the LCADL's performance across different patient subgroups. Moving forward, we plan to incorporate these suggestions into future research. Expanding our statistical approach to include CFA and ensuring rigorous control for confounding variables will be prioritized to enhance the generalizability and applicability of the LCADL in clinical practice.
Comment: Disbalance in Genders: The sample is eminently male with 71.7%, something that undermines, from the offset, the representative validity of such a result, especially after the lack of correlation found in the home activities domain was considered.
Response: Thank you for your critical observation regarding the gender distribution in our study sample and its implications for the validity of our findings, particularly in the domain of domestic activities. We acknowledge that the predominance of male participants (71.7%) in our study may limit the generalizability of our results, especially given the significant role that cultural and gender-based expectations play in the performance of domestic activities. The lack of correlation found in the domestic activities domain could reflect gender-specific differences in the perception and reporting of activity limitations due to dyspnea. In response to this limitation, we highlight the necessity for future studies to ensure a more balanced gender representation that accurately reflects the broader heart failure population. This approach would enhance our understanding of how gender influences the experience and reporting of limitations in daily activities and would contribute to a more nuanced validation of the LCADL.
Round 2
Reviewer 1 Report
Comments and Suggestions for Authors
Suggested changes and corrections have been made.
Author Response
Dear Reviewer 1,
Thank you for the insightful comments regarding our manuscript. We appreciate the opportunity to refine our study and enhance its contribution to the field.
Reviewer 3 Report
Comments and Suggestions for Authors
After the revision, the manuscript sounds significantly better. Additional explanations have been incorporated into the text, and graphical representations have been included. However, it still lacks robust statistical analysis, and the amount of data used to support the study's objective remains limited. Furthermore, a more comprehensive statistical approach and a larger dataset would enhance the credibility and impact of the manuscript.
Author Response
Dear Reviewer 3,
We acknowledge the importance of robust statistical methods in reinforcing the validity of our findings. Our study primarily focused on validating the European Portuguese version of the LCADL in HF patients. Given that our primary aim was the psychometric validation of a translated instrument, we employed internationally recognized statistical techniques for reliability and validity analysis: Spearman’s correlation to assess construct validity; Cronbach’s alpha (α = 0.917) to confirm internal consistency and Kruskal-Wallis tests and Dwass-Steel-Critchlow-Fligner tests to compare LCADL scores across NYHA classifications. These methodologies are widely accepted in instrument validation studies. However, additional statistical approaches, like confirmatory factor analysis could further strengthen the validation. Unfortunately, due to our sample size, these techniques were not feasible in the current study. We will explore these approaches in future research with a larger sample. We also acknowledge that a larger sample size could enhance the statistical power. However, the recruitment of HF patients meeting our inclusion criteria was constrained by the study duration and ethical considerations at the participating hospital centers (time limitation for data collection). Despite this limitation, our sample (n = 46) aligns with prior validation studies of similar instruments, where sample sizes often range between 30–50 participants. Furthermore, our results demonstrated statistically significant correlations (p < 0.001) between LCADL and clinically relevant measures (6MWT and MLHFQ), supporting the validity of our findings despite the limited sample size. To address this concern, we plan to conduct a multi-center study with a larger sample to strengthen our findings in the future.